# From Bench to Bedside: What Do We Know about Imidazothiazole Derivatives So Far?

**DOI:** 10.3390/molecules28135052

**Published:** 2023-06-28

**Authors:** Mu Guo, Xiangbin Yu, Yi Zhun Zhu, Yue Yu

**Affiliations:** 1School of Pharmacy, Fujian Medical University, Fuzhou 350122, China; guomu727@163.com (M.G.); yxb4666@fjmu.edu.cn (X.Y.); 2Fujian Center for New Drug Safety Evaluation, Fuzhou 350122, China; 3School of Pharmacy, Macau University of Science and Technology, Macau 999078, China

**Keywords:** condensed heterocyclic compounds, imidazothiazoles, pharmacological effect, druggable candidates, levamisole

## Abstract

Imidazothiazole derivatives are becoming increasingly important in therapeutic use due to their outstanding physiological activities. Recently, applying imidazothiazole as the core, researchers have synthesized a series of derivatives with biological effects such as antitumor, anti-infection, anti-inflammatory and antioxidant effects. In this review, we summarize the main pharmacological effects and pharmacological mechanisms of imidazothiazole derivates; the contents summarized herein are intended to advance the research and rational development of imidazothiazole-based drugs in the future.

## 1. Introduction

Condensed heterocyclic compounds are a series of compounds formed by condensing a benzene ring and a heterocyclic ring, or two heterocyclic rings together. These compounds have demonstrated potential physiological activities such as anticancer, anti-inflammatory, antioxidant, antiviral and antibacterial activities, making them highly promising for new drug development. In fact, many of these compounds have already been widely used in clinical practice [1,2].

Imidazole is a planar five-membered ring (Figure 1A, left) with two meta-nitrogen atoms, located at the 1st and 3rd position, which is a strongly polar compound with a dipole moment of 3.61D. Due to the presence of hexavalent π electrons, this compound is classified as an aromatic compound [3]. Furthermore, since the unshared electron pair of the 1-position nitrogen atom in the structure participates in the cyclic conjugation, the electron’s density therefore decreases. Compounds containing imidazole rings usually have verities of pharmacological activities such as antibacterial [4,5], antifungal [6], anticonvulsant [7], antioxidant [8], antitumor [9] and antiparasitic activities [10]. In terms of druggability, for example, for the exploration of various N-substituted imidazole inhibitors of p38 mitogen-activated protein kinases, it has been demonstrated that only substituents at the *N-* of the imidazole ring are tolerated without loss of activity, and appropriate substituents will help enhance the binding capacity of the compound to p38α [11]. In addition, it has been reported that the free nitrogen of the imidazole ring of the compound could produce water-mediated interactions with the side chain of Arg85 and the OH-4 of the GlcNAc moiety of Globo H [12].

Like imidazole, thiazole is also a nitrogen-containing, five-membered, heterocyclic aromatic hydrocarbon with a pyridine-like odor (Figure 1A, right), which has one nitrogen atom and one sulfur atom in its chemical structure. Meanwhile, thiazole is also an important skeleton for many synthetic compounds. Thiazole compounds can bind to various targets such as enzymes or receptors in the organism and exhibit various biological activities. Many biological activities of thiazole and bisthiazole derivatives have been reported [13], such as analgesia [14], anticancer [15,16], antioxidant [17], anti-inflammatory [18], antibacterial [19], and cardioprotective activities [20]. In these compounds, the thiazole ring is generally considered to be one of the important structures for its pharmacological activity, and it is also the structure required for the compounds to ensure druggability. Studies have reported the synthesis of a thiazole compound [21] with excellent performance in the fields of stability, solubility and synthesis. The positive charge of the compound is provided by the quaternary ammonium salt formed by the thiazole ring. At the same time, the hydrogen atom of the thiazole ring can interact with the oxygen atom of water through hydrogen bonds, and the sulfur atom of the thiazole ring will also contribute to the stability of the crystal structure.

In view of the pharmacological activities and structural characteristics of the two heterocycles, derivatives of imidazothiazole have been widely studied due to their extensive pharmacological effects and good biocompatibility, while compounds containing imidazothiazole structures have been demonstrated in clinical practice, exhibiting good clinical pharmacological activities, such as levamisole and quizartinib [22].

Generally, fusion of two heterocycles can integrate the biological properties of both, while the imidazothiazole ring is also a key active site and an important structure that can improve the stability of the compounds [23]. For example, in one study [24], the core of the imidazothiazole ring of the synthesized compound can form a nitrogen–iron bond with the heme group, which is one of the important ways for the compound molecule to bind to the target group. In another study, IT1t, a potent CXCR4 antagonist, with its imidazothiazole ring, can be linked by a short flexible linker, thus specifically binding to the helix VII of the chemokine receptor CXCR4, finally forming a salt bridge with Glu288 [25]. According to the positions of heteroatoms and unsaturated bonds, imidazothiazoles have various fusion modes (Figure 1B) [26], among which, the [2,1-b] configuration has been most studied due to its diverse pharmacological properties (Figure 1C). In this paper, the pharmacological activities of imidazothiazole derivatives as the core are reviewed (Table 1) in order to advance the research and rational development of imidazothiazole-based drugs in the future.

## 2. Bench Study: Exploration of the Pharmacological Activity of Imidazothiazole

### 2.1. Antitumor

Cancer has become one of the leading causes of death, and in recent years, the incidence and mortality of malignant tumors have been steadily increasing, with tens of millions of deaths reported worldwide, of which, cancer deaths in China account for one-third of the world’s death cases [57,58,59].

#### 2.1.1. Effect towards Lung Cancer

Lung cancer is the second most common cancer globally, with a high rate of morbidity and mortality. Despite progress in understanding the biology of lung cancer, early diagnosis, treatment options, and drug resistance mechanisms, patients with advanced disease have a dismal prognosis [60].

For lung cancer cells, the Shaik group synthesized a series of imidazothiazole derivatives through Huisgen 1,3-dipolar cycloaddition reaction [27]. One configuration of the compound (Figure 2A) exhibited the most potent antiproliferative effect on A549 human lung cancer cells, with an IC50 value of 0.92 µM compared to the IC50 of doxorubicin of 1.778 µM. The results also indicated that the addition of an electron donor group to the benzene ring connected to the imidazothiazole moiety increased the compound’s cytotoxicity. Compound **1** was found to inhibit tubulin polymerization and alter the microtubule network structure, resulting in G2/M cell cycle arrest. Using nocodazole as a control, the IC50 values of compound **1** and nocodazole were 1.43 µM and 1.59 µM, respectively. To gain insight into the possible mechanism, a molecular docking study of compound **1** was conducted. The results showed that the oxygen from the methoxy group on the benzene substituent of the imidazothiazole ring of compound **1** was hydrogen bonded to the NH of Gln11 amino acid residue of α-tubulin, and the sulfur atom of the core scaffold imidazole was weakly bonded to the NH of Leu255 amino acid residue of β-tubulin. Additionally, the carbonyl oxygen atom of the compound was bound to the NH of Asn258 of α-tubulin by hydrogen bond, and βCys241 was bound to the oxygen atom of dimethoxy group on the triazole subunit substituent by weak electrostatic interaction.

Furthermore, the Abdel-Maksoud group synthesized a series of compounds featuring imidazothiazole rings with arylsulfonamide termini, and the results indicated that compound **2** was the most promising analog (Figure 2B) [28]. Compared to the positive control drug Sorafenib (IC50 value of 2.51 µM), the IC50 value of this compound against non-small cell lung cancer cell NCI-H460 was 0.845 µM, and the compound exhibited high selectivity for cancer cells.

#### 2.1.2. Effect towards Melanoma

Melanoma (MM) is a malignant tumor caused by melanocytes and is considered the most aggressive and deadly form of skin cancer [61]. RAF kinases (ARAF, BRAF and CRAF) play a critical role in the activation of the MAPK signaling pathway [62]. The RAS-RAF-MEK-ERK pathway is a potential target for melanoma therapy. Among the RAF kinase subtypes, BRAF is the main activator of the MAPK signaling pathway [63]. V600E mutated B-RAF is over-activated and over-expressed in some melanoma patients [64]. V600E-BRAF and RAF1 (C-RAF) kinase inhibitors have been reported as antiproliferative agents against melanoma [65,66].

The Anbar group designed a series of imidazothiazole compounds containing pyrimidine rings, of which compound **3** had the most therapeutic potential (Figure 3A) [29]. Molecular docking, kinetic simulations and QSAR studies were conducted to study the binding modes. The results showed that there was a meta-hydroxyl group on the substituted phenyl ring group of the imidazothiazole core, and the imidazothiazole structure had a significant impact on the activity of this compound. Compound **3** had high potency and relative selectivity to V600E-B-RAF and was the only compound to execute a sub-nano IC50 value against V600E-B-RAF. Additionally, the compound displayed high efficacy against different melanoma cell lines with submicron molar IC50 values, and its selectivity for melanoma cells was higher than that of normal skin cells. Furthermore, it was able to penetrate the cell membrane and inhibit V600E-B-RAF kinase with an IC50 value of 0.19 μM, which was comparable to its IC50 value against UACC-62 melanoma cells (0.18 μM). Compound **4** is another imidazothiazole compound with high efficacy against melanoma (Figure 3B) [30]. The docking study revealed that compound **4** made molecular interactions with Asn 580 and Phe 583 amino acids and shared the same binding groups towards BRAF kinase enzyme active site with H-bonding interaction (sulfonamide chemical moieties) as well as arene–arene interaction (imidazolyl ring). The IC50 values of the compound on A375 and SK-MEL-28 MM cells were 7.04 µM and 9.40 µM, respectively, while the IC50 values of the control drug Sorafenib were 7.88 µM and 9.45 µM, respectively, indicating that compound **4** had a superior anti-MM cell effect.

#### 2.1.3. Effect towards Breast Cancer

Breast cancer is the most prevalent type of cancer among women globally and remains the second leading cause of cancer-related deaths in women [67]. The Shareef group designed and synthesized a series of aryl hydrazoles based on imidazothiazole and evaluated their antiproliferative potential on different human cancer cell lines [31]. Among the synthesized compounds, compound **5** (Figure 4A) had an anti-breast cancer cell line MDAMB-231 effect, with an IC50 value of 1.65 μM. Cell cycle analysis showed that the compound could make MDA-MB-231 cells stagnate at the G0/G1 phase and induce apoptosis of MDA-MB-213 cells. Furthermore, detailed biological studies, such as the Annexin V-FITC assay, DAPI staining, DCFH-DA assay and JC-1 staining, suggested that this compound has the potential to induce apoptosis of breast cancer cells. The Bin Sayeed group synthesized new compounds with an imidazothiazole backbone, with tubulin polymerization inhibition effect [32]. Their antiproliferative activity against different cancer cell lines such as breast cancer cell lines (MDA-MB-231 and MCF-7) was investigated in experiments, and the results showed that compound **6** with quinoline substitution (Figure 4B) was the most effective among them, with an IC50 value of 4.6 μM. Cell cycle assays showed that cell growth was arrested at the G2/M phase of the cell cycle, which led to apoptosis and cell death.

#### 2.1.4. Effect towards Other Tumors

The Baig group synthesized a series of imidazothiazole benzimidazole conjugates and evaluated their anti-proliferation activities against two human cancer cell lines, such as HeLa cervical cancer cells and DU-145 prostate cancer cells [33]. Taking the antineoplastic drug nocodazole as the positive control, the results showed that compound **7** (Figure 5A) had a similar anti-prostate cancer cell effect as the nocodazole. In addition, the results of cell cycle experiments showed that the compound could block the cell cycle in the G2/M phase, and the coupling compound had a significant inhibitory effect on the assembly of tubulin in cancer cells. In order to further explore the mechanism of action between such compounds and tubulin, the Baig group conducted a molecular docking study, finding that compound **7** bound at the interface of *α*- and *β*-tubulin. The interaction was strongly stabilized by two hydrogen bonds, the first one by imidazo[2,1-b]thiazole with ASNα101 at distance of 2.31 Å. The second formed between benzimidazole NH and Tyrα224 at distance of 2.01 Å. The docking results showed that compound **7** was combined with amino acids of Ala12, Ser178, Thr179, Ala180 and Gln183 of α-tubulin and the amino acids of Leu248, Lys254, Leu255, Asn258, Met325 and Lys 352 of β-tubulin in a hydrophobic effect.

Karaman and Ulusoy Güzeldemirci synthesized a series of imidazothiazole derivatives, among which compound **8** (Figure 5B) containing 2-hydroxyphenyl substituent had extensive growth inhibitory activity on several cancer cells [34]. The compound had the most significant inhibitory activity on two leukemia cell lines, SR and HL-60, with growth inhibition rates of 111.51% and 103.50%, respectively, and the inhibition rate on prostate cancer cell line DU-145 reached 108.65%. Compared with cisplatin and sorafenib, compound **8** showed a certain degree of inhibitory activity on most tumors. For OVCAR-3 ovarian cancer cells, compound **8** showed a better growth inhibition curve than the control drug. In addition, compound **8** also inhibited CCRF-CEM cells, HCT-15 colon cancer cells and CNS (SF-268) cell lines to some extent.

### 2.2. Anti-Infection

Pathogens such as viruses, bacteria and parasites have become a major public health concern, leading to dangerous infectious diseases and a high mortality rate in recent decades [68,69,70,71]. Commonly used anti-infection drugs include antibiotics, sulfonamides and quinolones, while imidazothiazole compounds have been shown to possess anti-infective properties.

#### 2.2.1. Antiviral Activity

Coronavirus disease 2019 (COVID-19) is a global infectious disease caused by severe acute respiratory coronavirus 2 (SARS-CoV-2). At present, few approved preventive or therapeutic drugs can be used for COVID-19. Some therapeutic agents are used alone or in combination. Compound **9** is a kind of drug with an imidazothiazole ring (Figure 6A) called levamisole, which has immunomodulatory effects. The immunological effects of levamisole can be achieved by improving T cell functions, regulating B and T cell functions and antibody productions, activating IL-18 to improve immune response, and activating monocyte–macrophage axis functions [35]. Therefore, considering the pathophysiology of COVID-19 and the immunomodulatory properties of levamisole, a double-blind randomized controlled trial was conducted to evaluate the effectiveness and safety of levamisole compared with conventional nursing standards in non-hospitalized patients with mild to moderate COVID-19. The results showed that levamisole could effectively improve some clinical conditions compared with placebo [36]. Another study also showed that levamisole could be effectively used for antiviral treatment, especially in COVID-19 patients with diarrhea [37]. In addition, levamisole has bidirectional effect in the treatment of COVID-19. In the early stage of COVID-19, it has a strong immune stimulation effect by regulating the cellular and humoral immune response, so as to promote the clearance of SARS-CoV-2 and tissue repair. However, it exerts harmful effects in the late phase of COVID-19 through suppression of Th2 immune response and propagation of proinflammatory cytokines to cytokine storm (Figure 6D) [35].

Coxsackievirus is a human pathogen, belonging to the *enterovirus* genus. The central nervous system of newborns is the main part of its infection, with high morbidity and mortality [72]. The Gürsoy group synthesized an imidazothiazole analog (Figure 6B), and its therapeutical effect towards Coxsackie B4 virus was investigated [38]. Compared with ganciclovir as reference, compound **10** had better therapeutical effects against feline coronavirus and feline herpesvirus. *Argentinian mammarenavirus*, better known as the *Junin virus* or *Junín virus* (JUNV), is an arenavirus in the *Mammarenavirus* genus that causes Argentine hemorrhagic fever (AHF). JUNV seriously threatens public health and safety and caused more than 5 million people to be trapped in the threat of disease [73]. The Barradas group designed and synthesized a series of 3,5-disubstituted imidazothiazole compounds, and their cytotoxicity and antiviral activity were then evaluated (Figure 6C) [39]. Results showed compound **11** and **12** had better antiviral activity than ribavirin in monkey Vero cells. In addition, compound **11** could also play an antiviral role in the cellular proliferation cycle of JUNV, and it was also an effective inhibitor of JUNV in human cells.

#### 2.2.2. Antibacterial Activity

Antibacterial agents are compounds that can kill or inhibit the growth of microorganisms, and the discovery and research of antibacterial drugs, particularly antibiotics, have significantly improved quality of life and increased life expectancy [74]. Currently, the literature reports classify antibacterial drugs into β-lactamides, aminoglycosides, quinolones, macrolides, glycopeptides, tetracyclines, and more. However, further research has revealed other chemical structures with similar antibacterial activity, such as derivatives containing imidazothiazole structures.

Tuberculosis caused by Mycobacterium tuberculosis (MTB) is a chronic but fatal infectious disease and one of the major causes of death from infectious diseases worldwide [75]. Imidazothiazole derivatives have attracted much attention in the last decade and might become potential therapeutic drugs. Ulusoy and Gürsoy synthesized a class of imidazothiazole derivatives, and the BACTEC460 radiation measurement system was used to evaluate the in vitro anti-mycobacterial activity of the synthesized compounds against Mycobacterium tuberculosis H_37_R_V_ [40]. These compounds showed different degrees of inhibition in in vitro screening at the same concentration, while compound **13** (Figure 7A) had the best in vitro activity, with a MIC of 6.25 μg·mL^−1^. In addition, the compound was also tested for cytotoxicity (IC50) in VERO cells at concentrations equal to the MIC for Mycobacterium tuberculosis H_37_R_V_. Then, the IC50 value was found at a concentration level of >10 μg·mL^−1^ for compound **13,** and the resulting selectivity index (SI= IC50/MIC) was calculated as >1.6, showing that this compound not only displayed a considerable antimycobacterial activity, but also had remarkable cytotoxicity. The Syed group used 2-aminothiazole as the starting material to synthesize a series of new imidazothiazole compounds by a multi-step method and screened all compounds for their anti-tuberculosis and antifungal activities [41]. The MIC of compound **14** (Figure 7B) against conjugated bacteria was 1.6 μg·mL^−1^, showing good antituberculosis effect, and compound **14** also showed ideal antifungal activity, with MIC of 25 μg·mL^−1^. Molecular docking studies were also carried out. The results showed that pantothenic acid synthase, enoyl acyl carrier protein reductase and aminoglycoside 2′-N-acetyltransferase of Mycobacterium tuberculosis was the effective target of anti-tuberculosis drugs. The Moraski group found and synthesized imidazothiazole-5-formamide compounds (compound **15** and **16**, Figure 7C,D) [42]. These compounds were a new class of anti-tuberculosis compounds with excellent activity against replicating and the ability to reduce bacterial burden in macrophages, which could fight against the drug resistance of Mycobacterium tuberculosis and have low toxicity to VERO cells with MIC value lower than 10 nM. The targeting selectivity of this series of compounds was then confirmed by cross resistance of specific QcrB mutants and high susceptibility to mutants with cytochrome bd oxidase function gene deletion.

At present, the infection caused by methicillin resistant Staphylococcus aureus (MRSA) is almost all over the world, and it is one of the most dangerous pathogens in hospitals, leading to many infections and even deaths [76]. The Li group synthesized a series of 5,6-dihydroimidazothiazole compounds, purified thorough HPLC, and investigated their antibacterial activity against the MRSA strain [2]. In this experiment, the multidrug resistant BAA-44, which was resistant to almost all kinds of antibiotics, including β-lactamides, macrolides, tetracyclines and fluoroquinolones, was used, while compound 17 (Figure 7E) showed the best antibacterial effect in this study. In addition, the experimental results also showed that the MIC90 value of R enantiomer was about half of its corresponding S enantiomer value, suggesting that the R configuration of the 6-position imidazothiazole scaffold had increased the antibacterial activity towards MRSA. DNA cyclooxygenase is essential in all bacteria, but it does not exist in higher eukaryotes. In view of the structural similarity between these imidazothiazole compounds and the recently reported benzimidazole urea compounds as inhibitors of DNA gyrase, the compound showed inhibition of DNA gyrase activity in the determination of the DNA gyrase supercoil assay, indicating that the compound might target DNA cyclooxygenase and play an antibacterial role.

The Shareef group designed and synthesized a triazole imidazothiazole hybrid and evaluated its antibacterial activity [43]. Among them, compound **18** (Figure 7F) had broad-spectrum antibacterial activity, and field emission scanning electron microscopy showed that the compound also showed significant anti-biofilm activity, which could destroy the single biofilm and mix biofilm of bacteria. In addition, molecular docking studies showed that they could interact with the virulence factor *Staphylococcus aureus* dehydrogenase quinoline synthase. Then, compound **18** exhibited lower cytotoxicity to normal cell lines compared to the antibacterial activity and showed non-toxic activity to MRC5 normal cell line.

#### 2.2.3. Anti-Parasitic Activity

*Acanthamoeba castellanii (A. castellanii)* of the T4 genotype is a causative agent of Acanthamoeba keratitis (AK), a blinding eye infection [77]. The Akbar group investigated the antiamoebic activity of several imidazothiazole derivatives against *A. castellanii* of the T4 genotype [44]. The Trypanosoma blue exclusion test and haemocytometer counting were used to assess the reduction in *A. castellanii* trophozoite proliferation following treatment with these compounds. Compound **19** (Figure 8A) and **20** (Figure 8B) exhibited the highest amoebicidal effects, eliminating 67% and 70% of *A. castellanii*, respectively. Furthermore, compounds **19** and **20** were able to block the binding of 55% and 61% amoeba to human cells, respectively, with minimal cytotoxicity to host cells and a significant reduction in amoeba-mediated host cell death.

Levamisole not only has an immunomodulatory effect, but also showed its antiparasitic effect both in vitro and in vivo. In Okolie’s study, thirty patients infected with ascaris lumbricoides and thirty patients infected with hookworm were treated with a clinically accepted duration of levamisole. The stool samples before and after treatment were examined microscopically for ova of Ascaris lumbricoides and hookworm and for total egg counts. The cure rate of ascaris lumbricoides infection and hookworm infection was 73.7% and 66.7%, respectively, taking no helminth eggs in stool samples after treatment as the cure standard [78]. In the course of anthelmintic treatment, sometimes the clinical treatment method of combined medication is adopted. Anto and Nugraha conducted a double-blind randomized clinical trial to compare the efficacy and side effects of albendazole, albendazole combined with levamisole and mebendazole combined with levamisole in the treatment of trichomoniasis [45]. The results showed that the cure rate of albendazole combined with levamisole for mild caterpillar disease and mixed infection was better, and the side effects of all treatment groups were similar.

### 2.3. Anti-Inflammatory

Inflammation is an immune response to invading pathogens and a defense mechanism against external stimuli, characterized by redness, swelling, pain and fever. Additionally, research has linked inflammation to a variety of life-threatening and debilitating diseases.

#### 2.3.1. Cellular Level Anti-Inflammatory Effect

The Soyer Can group synthesized a series of new formamide compounds with imidazothiazole as the skeleton [46]. In order to determine the anti-inflammatory activity of the compound on RAW 264.7 cells, a Griess test was conducted [79]. Nitrite determination showed that compound **21** (Figure 9A) had certain anti-inflammatory activity. In addition, compound **21** has been tested at 1.5 μM as the highest soluble concentration in DMSO, with a non-cytotoxicity. The Powers group found that if there was a compound with a polar substituent at position 2 or 3 of the imidazothiazole ring, the compound would maintain its anti-inflammatory activity and reduce its acute toxicity [47]. The addition of gem dimethyl substituent at position 6 would increase acute toxicity and lose anti-inflammatory activity. However, alkyl sulfonyl substituents were introduced into the thiazole ring of compound **22** (Figure 9B), making it have the best anti-inflammatory activity and toxicity ratio.

N-formyl-methionyl leucyl phenylalanine (FMLP) and other N-formylpeptides are activators of polymorphonuclear and mononuclear phagocytes. They can cause cell polarization, production of reactive oxygen species, production of arachidonic acid metabolites and release of lysosomal enzymes in granulocytes [80]. Neutrophils are important defensive components of the immune system and participate in the uptake and degradation of microorganisms. However, in some cases, neutrophils seem to be inappropriately activated to release tissue damaging molecules (such as protease) or molecules that can promote inflammation, thus leading to the deepening of inflammation. There is also evidence in animal models that neutrophils can promote the initiation and development of arthritis models [81]. The Andreani group synthesized a class of derivatives with an inhibitory effect on human neutrophils, which could be considered as potential anti-inflammatory agents [48]. Among which, after cell pulse, compound **23** (Figure 9C) could inhibit the lysozyme degranulation induced by the receptor agonist FMLP as well as chemotaxis activated by FMLP. This kind of compound could prevent neutrophils from recruiting to the inflammatory lesion site, showing a protective effect against direct damage to target tissues caused by the release of related enzymes.

#### 2.3.2. Paw Edema Mitigation

The rat paw edema model induced by carrageenan is often used to evaluate or screen new anti-inflammatory compounds, and injection of carrageenan into the underside of the rat paw will cause biphasic edema [82,83]. The Shetty group evaluated the anti-inflammatory activity of their synthesized new imidazothiazole sulfide and sulfone compound [49]. The results showed that compound **24** (Figure 9D) with *Br-* at the 4th position on the benzene substituent had the best anti-inflammatory activity. The results also showed that when the 4th position was substituted by *N-* or *Cl-*, the anti-inflammatory activity would decline, indicating that different substituents on the ring would have different effects on the anti-inflammatory activity.

#### 2.3.3. Neuroprotection Activity

The process of neuroinflammation is related to the pathogenesis of many cardiovascular diseases and plays an important role in extracerebral diseases, including heart diseases [84,85]. The Leoni group synthesized 4-imidazothiazole-1,4-dihydropyridine (compound **25**, Figure 9E) and found that the substituents on C-2 and C-6 of the bicyclic scaffold could affect cardiovascular parameters and play a neuroprotective role [50]. When compared with nifedipine, compound **25** showed better neuroprotective properties. Therefore, the appropriately decorated 1,4-DHP scaffold might be used for the modulation of brain calcium channels. However, compared with other substituents systems, compounds with a 1,4-DHP scaffold bearing in C4 a differently substituted imidazo[2,1-b]thiazole system might show the better neuroprotective effect and cardiovascular activity.

#### 2.3.4. Gastrointestinal Protection

Non-steroidal anti-inflammatory drugs (NSAIDs) are still the most-used treatment for rheumatic diseases. However, NSAIDs could lead to serious adverse reactions, the most serious of which is stomach damage, even gastric ulcer and kidney damage. Prostaglandin series bioactive compounds are formed by the interaction of two different but related enzymes, cyclooxygenase-1 (COX-1) and cyclooxygenase-2 (COX-2). Both subtypes contribute to the inflammatory process, but COX-2 has considerable therapeutic significance, because, during acute and chronic inflammation, the formation of prostaglandins will be enhanced after induction. Therefore, while maintaining anti-inflammatory activity, selective COX-2 inhibitors can also reduce gastrointestinal adverse reactions due to avoiding the synthesis of mucosal prostaglandins. Selective COX-2 inhibitors may become an important supplement to analgesic and anti-inflammatory drugs [86,87].

The Shahrasbi group synthesized imidazothiazole analogues containing methyl sulfonyl COX-2 pharmacophore and evaluated their COX-2 inhibitory activity [51]. According to COX-1/COX-2 inhibition data in vitro, all compounds were proved to be the selective inhibitors of COX-2 isoenzymes. However, according to the results, the effectiveness and selectivity of COX-2 inhibition activity were affected by the type and size of amine on the C-5 of imidazothiazole ring. When the C-5 was a disubstituted amino group (Figure 9F), compound **26** had the most effective selective COX-2 inhibition. In addition, the interaction between compound **26** and the COX-2 binding site was evaluated by molecular docking experiments. The results showed that the p-SO2Me substituent insertion existed in the secondary pocket of COX-2 (Arg 513, Phe 518, Val 523) and formed a hydrogen bond between the O atom of SO2Me and the amino group of Arg 513. At the same time, additional docking studies showed that all structures of compound **26**, including the imidazothiazole ring, closely overlapped with the COX-2 inhibitor SC558 at the COX active site (Figure 9G).

### 2.4. Antioxidant Activity

The body will inevitably produce or obtain free radicals from the outside world. Excessive free radicals will lead to various diseases, such as cardiovascular disease, aging, cancer and so on [88]. DPPH radical scavenging is a fast, simple, cheap and widely used method to evaluate the antioxidant activity. A large amount of the literature has used this method to evaluate the antioxidant capacity of compounds [89].

It has been reported that some imidazothiazole derivatives have potential antioxidant activity. The Dincel group synthesized 1,2,4-triazol-3-thione imidazothiazole derivatives, and their antioxidant activities were evaluated by DPPH radical scavenging activity [52]. The results showed that compounds **27**, **28**, **29** and **30** (Figure 10A) all had antioxidant activity, which was equivalent to quercetin as a reference compound. At the same time, N-ethyl substitution would reduce DPPH radical scavenging activity. Docking studies were conducted to consider the possible binding modes of compounds; the imidazo[2,1-b]thiazole moiety played a significant role. The results showed that the imidazo[2,1-b]thiazole moiety of compound **27** formed a pi–cation interaction with ARG 127, Imidazo[2,1-b]thiazole moiety of compound **28** formed hydrogen bond interaction with LYS 49, and the imidazo[2,1-b]thiazole moiety of compound **29** formed a pi–pi stacking interaction with PHE 120. In addition, the molecular dynamic (MD) simulations showed that all molecules were highly stable during the MD simulation, while the RMSD analyses also proved that these compounds displayed structural stability during the MD simulations. The Andreani group synthesized imidazothiazole series of compounds with phenolic hydroxyl groups, among which, compound **31** (Figure 10B) had the best antioxidant activity, which was similar to that of quercetin [53]. In a further study, compound **31** could also inhibit the production of nitrite and the activity of NFκB, enhance the activity of QR1 and inhibit the activity of aromatase.

## 3. Bedside Study: The Distinguished Clinically Pharmacological Activity of Imidazothiazoles Makes It an Excellent Druggable Candidate

Imidazole-containing drugs are widely used in clinical medicine, such as metronidazole, ornidazole and cimetidine. These drugs are effective against dermatomycosis caused by yeasts, molds and anaerobic bacteria, as well as systemic and local infections caused by parasitic protozoa. Drugs containing thiazole rings, such as ceftazidime for antibiotics, lusutrombopag for thrombocytopenia, tenonitrozole for trichomoniasis, and dasatinib for antitumor, are also widely used in clinical settings. The pharmaceutical properties of compounds containing imidazothiazole structures have been demonstrated in clinical practice, exhibiting good clinical pharmacological activities.

### 3.1. Levamisole

Levamisole, an imidazothiazole-based drug, is the most widely used drug for a variety of applications. Initially developed in the mid-1960s as a veterinary drug to treat parasitic infections, further research and exploration revealed that levamisole could also be used for adjuvant treatment of cancer and adjuvant treatment after cancer chemotherapy, as well as the treatment of autoimmune diseases such as rheumatoid arthritis and lupus erythematosus [90]. Levamisole has been found to regulate and restore immunity, promote the differentiation and proliferation of T cells and induce the body to secrete cytokines such as IL-2. Additionally, it can strengthen the phagocytosis and chemotaxis of macrophages, improve the activity of natural killer cells, improve the immunity of the body, and restore the normal immune function of people with low immunity but has no obvious effect on the normal body [91,92]. In order to improve patient compliance, various forms of levamisole have been developed for oral administration, such as orodispersible tablets of levamisole hydrochloride or chewable tablets of levamisole [93,94]. Gastroretentive tablets of levamisole have been developed to prolong the gastric residence of levamisole or play the role of slow release in the stomach by a bioadhesive delivery system or stomach-floated drug delivery system [95,96]. Film-coated tablets of levamisole have been developed to obtain good taste-masking characteristics [97]. Additionally, Mianyikang^®^ (Fuzhou Chenxing Pharmaceutical Co., Ltd., Fuzhou, China) developed a liniment with levamisole as API, which shows a promising anti-allergic rhinitis effect.

Levamisole is a weak base, with a pKa value reported to be between 6.75 and 6.98. In the fasted stomach, the ionized form of levamisole is the predominant species, while in the small intestine, approximately half of the drug is present as the non-ionized form. The predicted log *p*-value of levamisole is 2.36 [98,99]. In one study, the plasma and urine concentrations of levamisole were measured in 10 healthy volunteers (including seven men and three women) after a single dose of 150 mg levamisole. Results showed that levamisole was rapidly absorbed (Tmax is 1.5 h), with a peak plasma concentration of 716.7 ± 217.5 ng·mL^−1^. The plasma elimination half-life of levamisole was 5.6 ± 2.5 h, and only 3.2 ± 2.9% of the oral dose returned to the unchanged drug in urine [100]. In another experiment, 11 healthy volunteers (six men, five women) and 12 cancer patients (six women, six men) were given a single dose of levamisole (2.5 mg·kg^−1^ body weight) by oral administration, while another group of the same experimental population was given a single dose of levamisole (5 mg·kg^−1^ body weight) by oral administration. The results showed that the drugs were well absorbed, with peak levels of levamisole in plasma reached after 1.0–2.0 h and ranging between 0.8 μg·mL^−1^ and 1.6 μg·mL^−1^. The absorption fractions were 0.7, and the clearance rate was 310 mL·min^−1^. The plasma half-life of levamisole varied between 3.3 h and 5.1 h. Twenty-four hours after administration, the concentration of levamisole in plasma was only six of the peak levels [101].

Levamisole is mainly metabolized by the liver, forming a variety of metabolites which are quickly eliminated. Most of the dose is excreted in the form of metabolites through urine, with only 3.17% of the dose recovered as levamisole. A small portion of the drug is excreted through feces. P-hydroxylation is a significant metabolic pathway of levamisole, with the p-hydroxylated metabolite mainly excreted in conjugation with glucuronic acid. Urine products include a small amount of levamisole prototype and free and bound p-hydroxy levamisole [100]. Interindividual variability and otherness in pharmacokinetic parameters are high, with the coefficient of variation for the total clearance of levamisole in healthy or cancer subjects being 36–38% or 36–37% [101]. Additionally, the absorption rate of levamisole is approximately twice as rapid in women as in men, resulting in higher and earlier peak concentrations and higher bioavailability than in male subjects [100]. Levamisole is generally well-tolerated at treatment concentrations; however, adverse reactions such as nausea, vomiting, headache, abdominal pain, myalgia, diarrhea, insomnia, blurred vision, dizziness, fatigue, and other common adverse reactions have been reported [102,103]. Additionally, immune-related adverse reactions such as decreased or absent white blood cells, particularly granulocytes, thrombocytopenia and purpura rash, have been observed [104,105].

### 3.2. Quizartinib

Quizartinib, also known as AC220, is a small molecule receptor tyrosine kinase inhibitor, originally from Ambit Biosciences and later acquired by Daiichi Sankyo, that is currently under development for the treatment of acute myeloid leukemia. Specifically, quizartinib selectively inhibits class III receptor tyrosine kinases, including FMS-related tyrosine kinase 3 (FLT3/STK1), colony-stimulating factor 1 receptor (CSF1R/FMS), stem cell factor receptor (SCFR/KIT), and platelet derived growth factor receptors (PDGFRs). Mutations cause constant activation of the FLT3 pathway resulting in inhibition of ligand-independent leukemic cell proliferation and apoptosis. Good results were obtained in a phase II clinical trial in 2012 for refractory AML, especially in patients undergoing stem cell transplantation [106].

Quizartinib is undergoing an open-label, multicenter Phase 2 clinical trial at 76 hospitals in the U.S., Europe and Canada. The trial included patients with pathologically diagnosed primary acute myeloid leukemia or acute myeloid leukemia secondary to myelodysplastic syndromes. A total of 333 patients were recruited, of which about 50% of FLT3-ITD-positive patients and about 30% of FLT3-ITD-negative patients achieved remission. The above results indicated that quizartinib is a rationally designed, orally administered, selective and potent FLT3 inhibitor. It has desirable pharmacokinetic characteristics with a long in vivo half-life and relatively low protein binding [107,108].

### 3.3. WAY-181187

WAY-181187 has demonstrated preclinical efficacy in rodent models of depression, anxiety, and notably obsessive–compulsive disorder, though it has also been shown to impair cognition and memory. WAY-181187 is a high affinity and selective 5-HT6 receptor full agonist. It induces robust increases in extracellular GABA levels in the frontal cortex, hippocampus, striatum and amygdala of rats, without affecting concentrations in the nucleus accumbens or thalamus, and has modest to no effects on norepinephrine, serotonin, dopamine or glutamate levels in these areas [109,110].

## 4. Conclusions

Numerous studies have demonstrated the pharmacological activities and therapeutic benefits of fused heterocyclic compounds, with imidazothiazole-containing compounds exhibiting great druggability due to their diverse pharmacological properties. Levamisole, a representative drug with an imidazo[2,1-b]thiazole ring, has been used clinically for decades, and further research is being conducted to uncover its additional pharmacological activities. As previously mentioned, Mianyikang^®^ (Fuzhou Chenxing Pharmaceutical Co., Ltd., Fuzhou, China) is a liniment containing levamisole hydrochloride as its main active ingredient and is used to treat diseases associated with low immune function and autoimmune diseases, such as recurrent upper respiratory tract infections, allergic asthma and allergic rhinitis. However, this article only provides an overview of pre-clinical trials. Further research is needed to prove the clinical safety of these compounds, as many of them are still in the experimental stage. Additionally, the instability of their synthesis must be addressed. Furthermore, the imidazothiazole ring has multiple conformations, and the imidazo[2,1-b]thiazole ring has been the most studied, suggesting that conformations play an important role in the study of pharmacological activities. Reports have also revealed that different substituents on the imidazothiazole ring can either enhance or reduce its pharmacological activity, indicating that selecting suitable substituents according to specific goals can maximize efficiency. Finally, while many articles have conducted simple cytotoxicity experiments in vitro, more comprehensive and systematic toxicological experiments are needed to assess the toxicity of these compounds.

In summary, this article reviewed the pharmacological activities of imidazothiazole derivatives and their clinical applications. These imidazothiazole-based compounds often demonstrate superior therapeutic effects compared to relevant control drugs. The clinical success of levamisole and its research potential further suggest that these compounds are worthy of further investigation. There is reason to believe that imidazothiazole derivatives possess great druggability, particularly derivatives of the imidazo[2,1-b]thiazole ring. Despite the encouraging results observed in many studies regarding their response to disease control, full characterization, such as toxicokinetic and pharmacokinetic studies, is recommended for their safe clinical use.

## Figures and Tables

**Figure 1 molecules-28-05052-f001:**
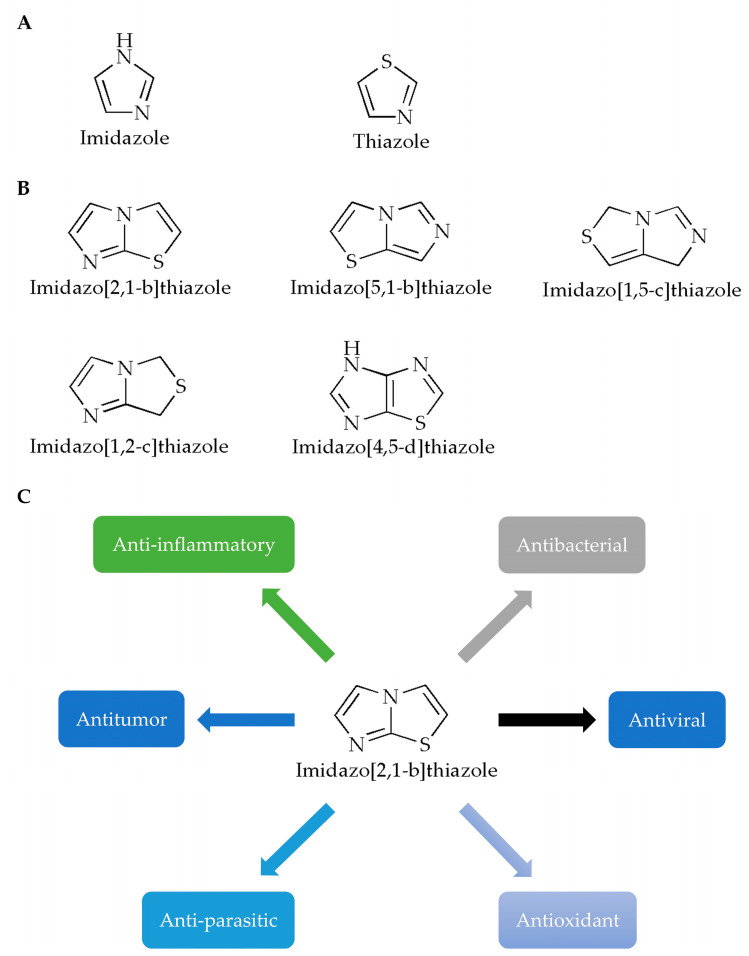
Chemical structures of (**A**) imidazole, thiazole and (**B**) imidazothiazole. (**C**) Studies on the pharmacological activity of imidazo[2,1-b]thiazoles.

**Figure 2 molecules-28-05052-f002:**
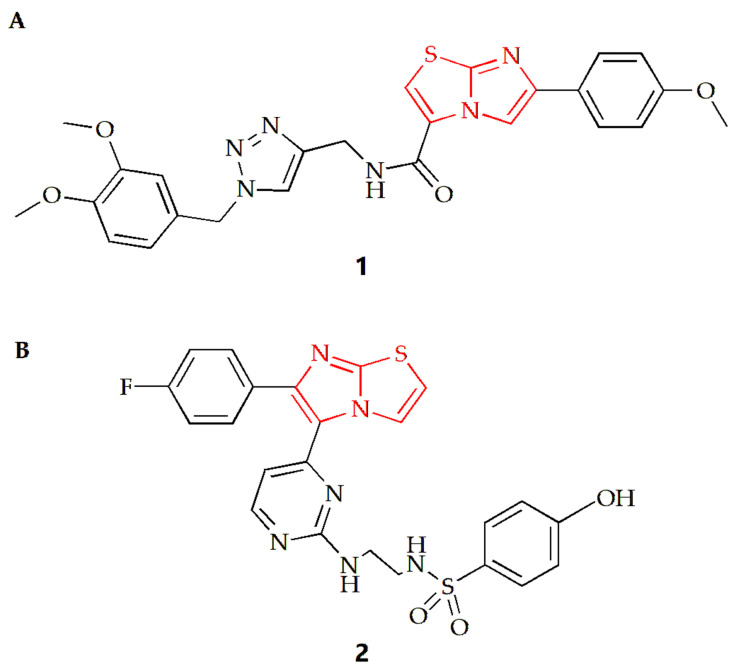
Chemical structures of (**A**) compound **1** and (**B**) compound **2**.

**Figure 3 molecules-28-05052-f003:**
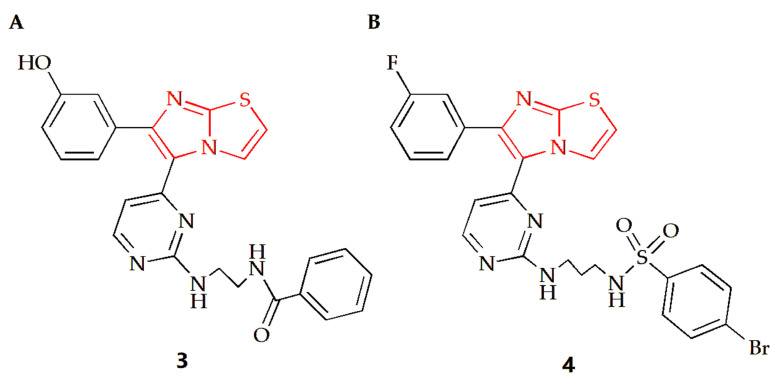
Chemical structures of (**A**) compound **3** and (**B**) compound **4**.

**Figure 4 molecules-28-05052-f004:**
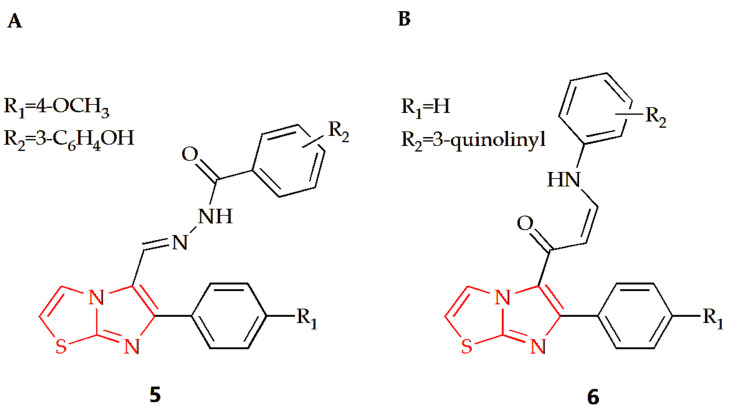
Chemical structures of (**A**) compound **5** and (**B**) compound **6**.

**Figure 5 molecules-28-05052-f005:**
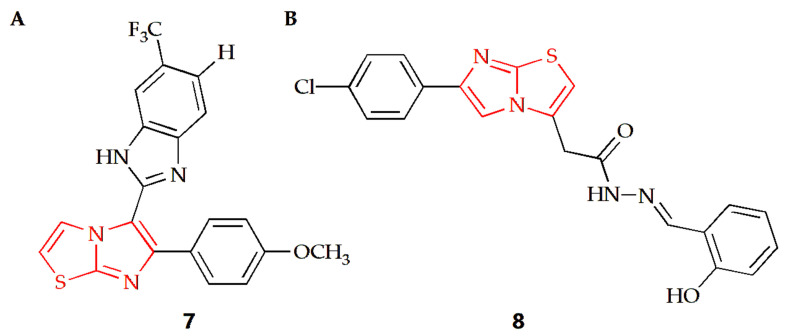
Chemical structures of (**A**) compound **7** and (**B**) compound **8**.

**Figure 6 molecules-28-05052-f006:**
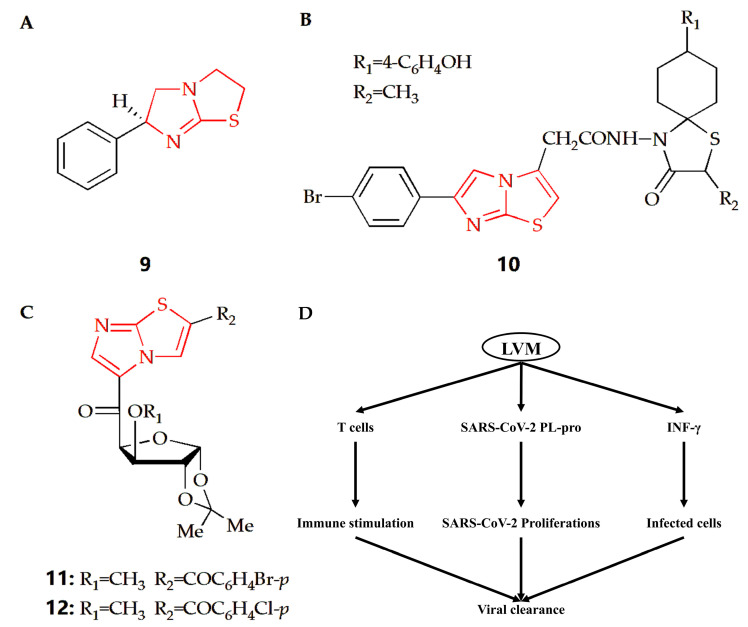
Chemical structures of (**A**) compound **9**, (**B**) compound **10**, (**C**) compound **11** and compound **12**. (**D**) The proposed potential levamisole treatment mechanisms in early SARS-CoV-2 infection. LVM, levamisole; PL-pro, papain-like protease.

**Figure 7 molecules-28-05052-f007:**
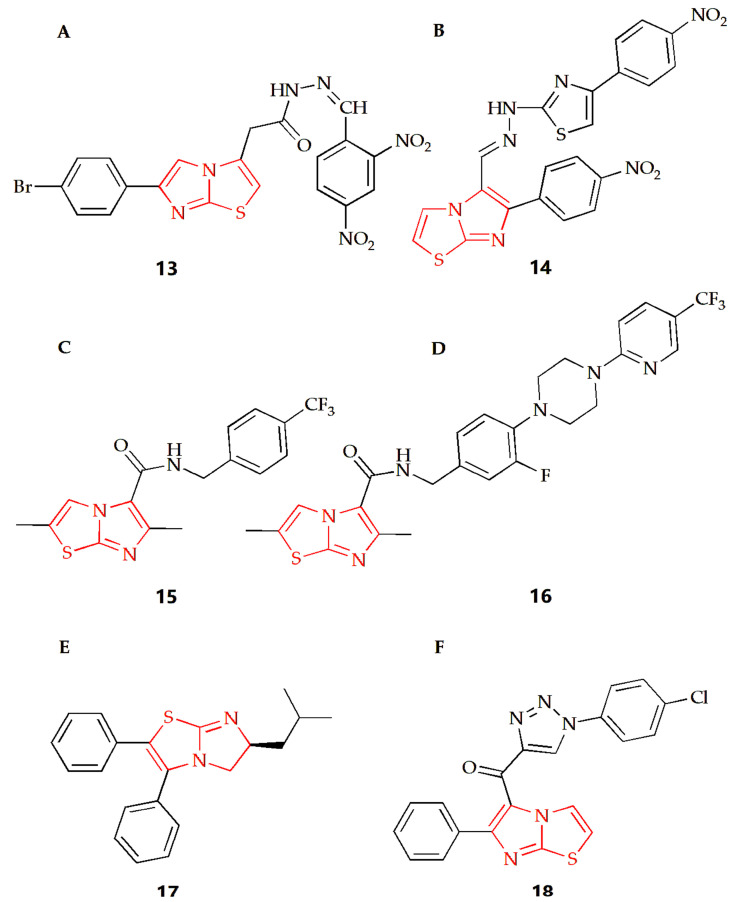
Chemical structures of (**A**) compound **13**, (**B**) compound **14**, (**C**) compound **15**, (**D**) compound **16**, (**E**) compound **17** and (**F**) compound **18**.

**Figure 8 molecules-28-05052-f008:**
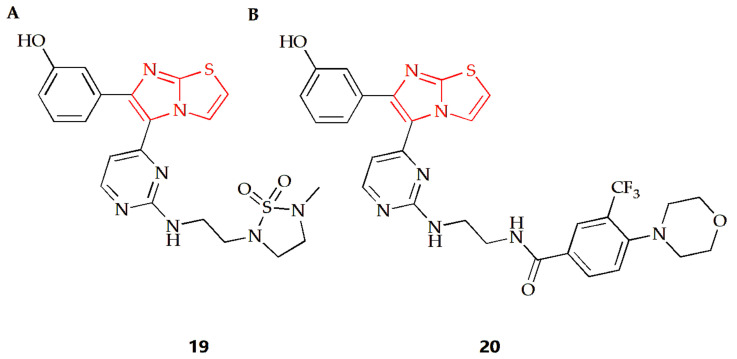
Chemical structures of (**A**) compound **19** and (**B**) compound **20**.

**Figure 9 molecules-28-05052-f009:**
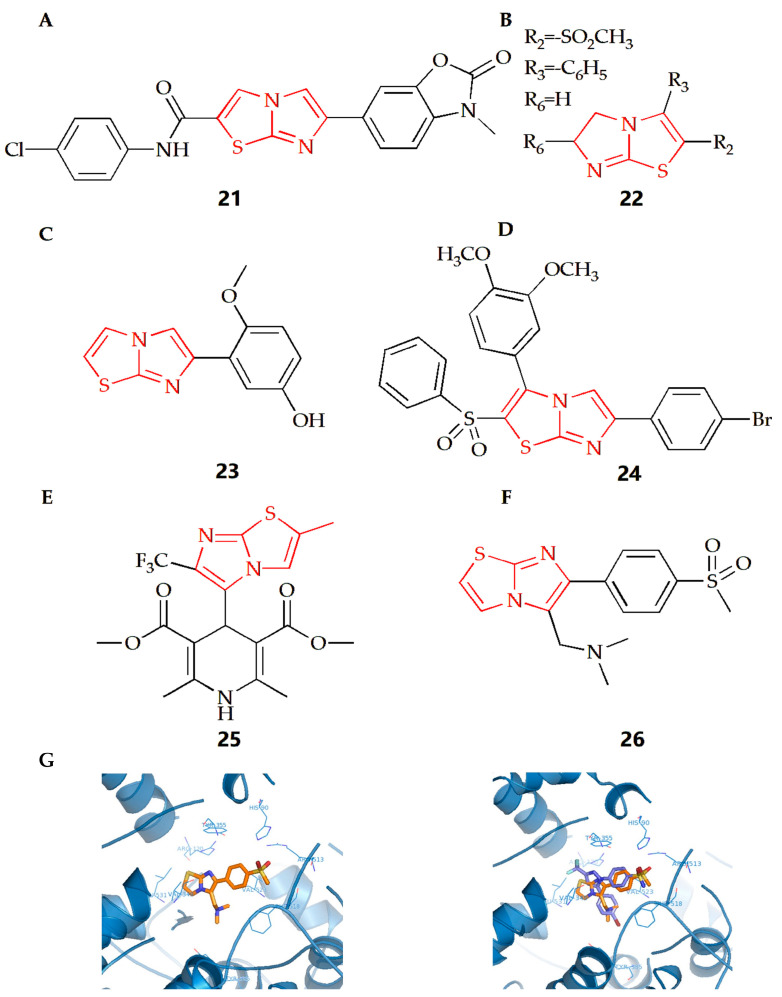
Chemical structures of (**A**) compound **21**, (**B**) compound **22**, (**C**) compound **23**, (**D**) compound **24**, (**E**) compound **25** and (**F**) compound **26**. (**G**) Compound **26** docked in the active site of murine COX-2. Hydrogen atoms of the amino acid residues have been removed to improve clarity (Left). Superimposition of compound **25** on the SC558 molecule (Right) (Adapted from [51]).

**Figure 10 molecules-28-05052-f010:**
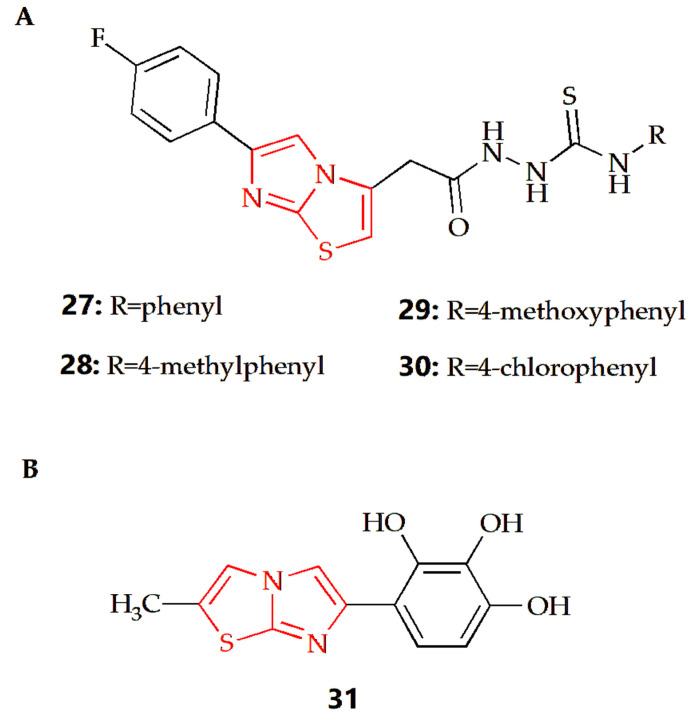
Chemical structures of (**A**) compound **27**, compound **28**, compound **29**, compound **30** and (**B**) compound **31**.

**Table 1 molecules-28-05052-t001:** Research of Pharmacological Activity of Imidazothiazoles.

Pharmacological Activity	Function Target	Ref.
Antitumor	Lung cancer	[27,28]
	Melanoma	[29,30]
	Breast cancer	[31,32]
	Other cancer	[33,34]
Anti-infection	Virus	[35,36,37,38,39]
	Bacteria	[2,40,41,42,43]
	Parasite	[44,45]
Anti-inflammatory	Inflammatory cells	[46,47,48]
	Paw edema model	[49]
	Nerves and cardiovascular system	[50]
	Cyclooxygenase-2 (COX-2)	[51]
Antioxidant activity	Excessive free radicals	[52,53]
Analgesic activity	Opioid µ-receptors	[54]
Therapeutic effect of cardiovascular disease	Voltage-gated Ca^2+^ channels	[55,56]

## Data Availability

Not applicable.

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
