# Peer review of "From Bench to Bedside: What Do We Know about Imidazothiazole Derivatives So Far?"

_molecules, 2023, doi:10.3390/molecules28135052_

Round 1
Reviewer 1 Report
Interesting work, well researched, based on reports from the last few years.In my opinion, it deserves to be published.
Author Response
Thanks for your kind suggestion and evaluations.
Reviewer 2 Report
Dear Authors
The present manuscript presents a review about the important pharmacological profile and clinical applications presented by imidazothiazole derivatives. It has blood for the area of ​​Medicinal Chemistry, as it summarizes important recent results, to be studied both in enzymatic assays and via Molecular docking studies. Consider this manuscript accepted after minor revision:
1. Appropriate references to the ACS format suggested by Molecules
2. Insert quotes below:
Muna Poudel; Garam Kim; Poshan Yugal Bhattarai; Seung Shin; Seyed-Omar Zaraei, Chang-Hyun OH; Hong Seok Choi. Potent Imidazothiazole-based Inhibitor of BRAF V600E Overcomes Acquired Resistance via Inhibition of RAF Dimerization in PLX4032-resistant Melanoma. Anticancer Research June 2022, 42 (6) 2911-2921; DOI: https://doi.org/10.21873/anticanres.15773
Mahmoud K. Shehata; Muhammad Uzair; Seyed–Omar Zaraei; Afnan I. Shahin; Syed J. A. Shah; Saif Ullah, Jamshed Iqbal; Mohammed I. El–Gamal. Synthesis, biological evaluation, and molecular modeling studies of a new series of imidazothiazole or imidazooxazole derivatives as inhibitors of ectonucleoside triphosphate diphosphohydrolases (NTPDases). Medicinal Chemistry Research (2023) 32:314–325. https://doi.org/10.1007/s00044-022-03000-y
Author Response
Thanks for your kind suggestion and evaluations. I have revised my manuscript under your guidance, please refer to the re-submitted manuscript.

Reviewer 3 Report
Review article “From Bench to Bedside: What Do We Know About Imidazothiazoles Derivatives So Far?” with authors Mu Guo, Xiangbin Yu, Yi Zhun Zhu, and Yue Yu is devoted to compounds with proven biological potential and to those that have been established in medical practice. The authors pay attention to the structural features of the compounds commented and to the influence of individual structural fragments on the biological activity shown. The authors refer to a large volume of cited literature from the last 20 years. The overview provided and the mentioned structure-activity relationship data would be useful in the future design of nоvel imidazothiazole-based compounds with expected specific biological properties and would facilitate and direct the choice of appropriate substituents and subunits into the main heterocycles.The data discussed by the authors are clean and well-presented. It is a well-written manuscript and qualifies for publication after minor revision. There are some comments and suggestions below need to be addressed before the acceptance of this manuscript
1) I believe that the inclusion of a figure with imidazole- or imidazothiazole-containing compounds with prominent pharmacological properties or already approved for clinical application would contribute the manuscript. This can be done in the Introduction.2) It would be better if the numbering of the compounds was in Bold-style. In present version, it is difficult to distinguished it from this of the cited literature.
3) Please match the size of the structural formulas of the compounds in all figures.
4) The substituents in Figure 4 are nonbold, while in all other cases (Figures 6 and 10, for example, are in a Bold typeface.5) Please revise all the cited literature according to journal requirements.
6) Оther remarks:
Line 93: A space between Huisgen and 1,3-dipolar cycloaddition is missing.
Line 96: Please change “an electron group” with “an electron donor group”.
Line 102: Please change “the methoxy oxygen” with “the oxygen from the methoxy group”.
Line 107: Please change “of dimethoxy” with “of dimethoxy group.
Line 173: There is an extra letter "c" before “nocodazole”.Line 186: Please add “substituent” after “2-hydroxyphenyl”.Figure 6C: Please modify substituents R2 – use small letter “p” and Italic style. It is about para-substitution.
Lines 246 and 284: “β-Lactamides” – Capitalization is not necessary.
Line 407: Please change “diamine substituent” with “disubstituted amino group”.

I have no special requirements regarding to the quality of English language.
Author Response
Thanks for your kind suggestion and evaluations.
- We have added some contents to introduction part, please review.
- Numbering of the compounds has been revised into Bold-style.
- Size has been modified.
- We have revised our careless mistake under your kind guidance.
- All the cited literature has been revised according to journal requirements.
- Relevant contents have been modified under your kind guidance.